# Production of Fish Analogues from Plant Proteins: Potential Strategies, Challenges, and Outlook

**DOI:** 10.3390/foods12030614

**Published:** 2023-02-01

**Authors:** Chengxuan Zhong, Yiming Feng, Yixiang Xu

**Affiliations:** 1Department of Agrotechnology and Food Science, Wageningen University & Research, 6708 PB Wageningen, The Netherlands; 2Department of Food Science & Nutrition, California Polytechnic State University, San Luis Obispo, CA 93407, USA; 3Healthy Processed Foods Research Unit, Western Regional Research Center, USDA-ARS 800 Buchanan Street, Albany, CA 94710, USA

**Keywords:** fish analogue, fish meat structure, plant protein, processing, sustainability

## Abstract

Fish products are consumed by human beings as a high-quality protein source. However, overfishing, and pollution puts out an urgent call to seek a new strategy to substitute fish protein for secure eco-sustainability. Plant-based fish analogs, which mimic the structure, texture, and flavor of fish meat products, are a rapid-growing segment of the food products. The purpose of this review is to discuss the feasibility and potential strategies for developing plant-based fish analog. The nutritional properties, especially the protein quality of plant-based fish analogs, were discussed. Furthermore, a thorough comparison was made between fish and terrestrial animal muscle structures, including both macroscopical and microscopical structures. Potential processing technologies for producing plant-based fish analogs from plant proteins and approaches for the characterization of the fish analog structures were elaborated. Comparing all the current processing techniques, extrusion is the predominately used technique in the current industry. At the same time, 3D-printing and electrospinning have shown the prominent potential of mimicking fish muscle structure as bottom-up approaches. Finally, key challenges and future research were discussed for the potential commercialization of plant-based fish analogues. The primary focus of this review covers the innovative works that were indexed in the Web of Science Core Collection in the past five years.

## 1. Present Global Seafood Production

According to the most recent data from the World Bank, the world population has increased more than twice in the past half century [1]. As one of the most important sources of high-quality protein, the demand for fish has increased drastically to meet the needs of the growing population. In the meanwhile, the average consumption per capita of fish has doubled in the past few decades, which in turn requires a fourfold increase in fish production [2]. Although fishery provides an important protein source for human being’s consumption, increasingly demanding fish will result in uncontrolled fishery or overfishing, and it will eventually surpass the rate of natural replenishment [3]. Over the past century, the number of the overfished region increased by two-fold globally, making one-third of the total world’s fisheries over biological limits [3]. According to the FAO, the recent increase in fish and seafood production, driven by aquaculture, made the capture fisheries reach a bottleneck in the early 1990s [4]. Furthermore, the current aquaculture practice also faces a few challenges, including (1) disruption of the marine ecosystem. For example, pelagic fish, commonly used as feeds, not only threaten the marine ecosystem, but also impede the growth of wild capture fisheries indirectly [5]. (2) Chemical use in aquaculture: antibiotics and pesticides, which are frequently used for disease management [6], will pollute the environment and impair human health if discharged into external water bodies [7]. (3) Effluent treatment: effluent that is generated during the culturing processes could also pose a huge threat to the environment. A recent study found that dissolved chemicals are more recalcitrant to degradation, and yet more cost-efficient treatments are still underway [7]. Therefore, aquaculture activities may lead to organic waste and toxic compound pollution, which post negative effects on the ambient aquatic ecosystem and reduce the productivity of the aquaculture [8]. In addition, other aquaculture-related pollution, such as plastic pollution or sunk fishing equipment, could also endanger the marine ecosystem [9].

Overfishing has multiple adverse effects on the ocean ecosystem, including an irreversible decline in fish stocks, maturation, and reproduction (e.g., bluefin tuna, Grand Banks cod) [3]. Several fishing strategies have been proposed and applied to prevent overfishing. For example, Individual Tradable Quotas (ITQs) are used to define the total allowable catch per boat, which aim to promote both higher profits for fishmen as well as relieving the pressure of the environmental ecosystem [10]. In the meanwhile, it is urgent for scientists to search for other alternative ways to alleviate the exploitation of the ocean ecosystem and secure sustainable growth. As a more sustainable solution shown in Figure 1, the plant-based fish analogue is proposed to tackle the ecological challenge. There have been many companies that successfully launched plant-based meat products on the market. However, to the best of the author’s knowledge, most of those products are meat analogues, while none have been found with fish analogues. Beyond Meat^®^ is known for their plant-based burger and minced meat, which is produced from soy protein and other plant proteins. Other startup companies also produce plant-based products with a great variety, such as the plant-based escalope and meatballs by Vegetarian Butcher^®^ and lupine-based egg pulp by Plan B^®^. With the rapidly diversifying categories of plant-based product innovation, it is anticipated that plant-based fish analogues will become trending in the near future. However, there is still a knowledge and technological gap to fill. Based on the existing research on meat analogues, the emulation of real fish fillet texture is difficult. Therefore, a comprehensive review of fish composition and structure is presented to reveal possible ways to mimic fish fillets at a micro-scale.

## 2. Major Composition of Fish

Fish is an aquatic organism with a great variety of species, and its chemical composition remarkably varies with different species. Water, protein, and fat account for up to 98% of the fish weight, and the weight ratio of major components depends on the fish species [8]. Tuna, cod, and salmon are recognized as the three major fish on the market [11] (Table 1).

### 2.1. Protein Composition

Muscle tissues take up more than half of total fish weight, and protein accounts for 16–21% of the major constituents [15]. The content of carbohydrates is relatively low, only comprising about 0.5% of the fish muscle [15]. There are three main groups of protein in fish muscle based on their solubility [16]: myofibrillar protein, sarcoplasmic protein, and stroma protein. Myofibrillar proteins, soluble in a concentrated salt solution, are the predominant group of proteins in fish muscle and account for 66–77% of the total mass of fish muscle protein. They consist mainly of myosin, actin, and other proteins, such as troponin and actinin, and can be classified as regulatory proteins and work as structural proteins [17]. Sarcoplasmic proteins, the second-largest group of proteins and the most soluble fraction in fish muscle, constitute 20 to 25% of the total fish muscle proteins [16]: they are a group of proteins that combine several water-soluble and low molecular weight proteins, such as albumins, myoglobin, hemoglobin, and enzymes [16]. Stroma proteins, including collagen and elastin, constitute connective tissue in fish muscle and are completely water insoluble [15,17].

### 2.2. Protein Quality

Protein Digestibility-Corrected Amino Score (PDCAAS), a rapid and routine method to determine protein quality, assesses the digestibility and amino acid score of a protein sample. PDCAAS is calculated as digestibility (x = mg of first limiting amino acid in 1 g of test protein/mg of the same amino acid in 1 g reference protein * 100%) [18]. PDCAAS values vary from 0 to 100%, and any value higher than 100% will be truncated to 100% [18]. As plant-based proteins are gaining increasing attention from the public, it is important to evaluate their protein quality to ensure that the nutritional properties of plant-based proteins are comparable to that of fish proteins. PDCAAS for some common plant and animal proteins are shown in Table 2.

PDCAAS of fish protein: Studies have shown that raw marine and freshwater fishes both have a high PDCAAS, which is affected negatively by different processing techniques [19,20]. For instance, broiling and smoking could lower the PDCAAS of rainbow trout [19]. Other common marine fish species, such as salmon, trout, and herring, all have PDCAAS of 1 after truncation [20]. A study of PDCAAS on some freshwater fish species found very high PDCAAS values [21].

PDCAAS of plant proteins: Many plants are known as good sources of proteins, including chickpeas, rice, lentils, and almonds. Soybean, one of the most common plants-protein sources for producing meat analogues, has a PDCAAS of 95% to 102% when untruncated depending on testing methods and soybean sources, mostly at a value of 100% after truncation [22,23]. Some plant protein sources have a slightly poorer PDCAAS, such as wheat, peas, and corn grain at levels of 46.3%, 78.2%, and 47.3%, respectively [22]. Soy protein, as a common source for plant-based meat, contains a relatively low methionine fraction, while many other plant-based proteins are rich in sulfur-containing amino acids [24,25]. Therefore, it could be a feasible solution to blend various plant proteins to increase their amino acid score by complementing limiting amino acids [24]. However, there are other factors that could limit the digestibility of plant protein, such as the antinutritional factors (e.g., Trypsin inhibitors in beans) found within many plants’ protein sources [24].

As shown in Table 2, soybean protein has a PDCAAS of 1 after truncation [23], which suggests that the amino acid composition of plant protein is not considered a limiting factor. However, the presence of anti-nutritional compounds in the plant protein could limit the bioavailability of protein, but they can be inactivated by procedures such as heat treatment and irradiation [26]. When comparing PDCAAS between fish and plant proteins, most fish species were found to have a higher untruncated PDCAAS value. However, it is a worthy note that one protein source with PDCAAS over 1 does not indicate that it is superior to its counterpart having PDCAAS at 1 if it is the sole source of protein in the diet [24]. It means that after truncation, fish protein PDCAAS does not show a superior characteristic to that of soy protein. As a result, the nutritional properties (PDCAAS) should not be considered a drawback when using plant-based protein for manufacturing fish fillet analogs.

**Table 2 foods-12-00614-t002:** PDCAAS (Protein Digestibility-Corrected Amino Score) for some typical animal and plant proteins.

Product Type	PDCAAS (after Truncation)	Reference
Cod	0.96	[27]
Tuna	0.97	[28]
Salmon	1	[20]
Herring	1	[20]
Mackerel	1	[20]
Trout	0.998	[19]
Beef	1	[22]
Casein	0.99	[28]
Soy protein	1	[23]
Chickpea protein	0.66	[28]
Lentils	0.52	[28]
Kidney beans	0.68	[28]

## 3. Fish Muscle Structure

To process plant-protein based fish analogue, it is firstly important to understand the natural structure of fish muscle. In fact, fish muscle presents a highly hierarchical structure, ranging from centimeters to nanometers (shown in Figure 2). Such a hieratical structure is associated with the unique texture, viscoelastic properties, and mouthfeel of fish products. This section aims to provide a detailed introduction to the fish muscle structure and alignment.

### 3.1. Fish Muscle Structure

Striated muscle and smooth muscle are the two main types of muscle in fish. Striated muscle, the major fraction of fish muscle, is characterized by transverse stripe and consists of white, red, and intermediate pink muscles with more than 90% of white muscle [30,31]. White muscle is characterized by its white to off-white color in most fish species [16], and it is the major constituent of the myotomal musculature in fish fillets with fiber diameter in the range of 50 to 100 μm (Figure 2) [31,32]. The red color in certain marine fish species, such as salmon and trout, is due to red carotenoids, rather than myoglobin [33]. From a functional point of view, since the white muscle works as sprinting muscle for prey to capture and escape from predators and is active at high cruising speed, it is also known as fast muscle [31,34]. On the other hand, among these three muscle types, red muscle presents the smallest fiber diameter in the range of 25 to 45 μm, but higher levels of lipids, haemoglobin, glycogen, and vitamins [33]. This protein is in the superficial area along the skin [35], and its major function is to provide slow and continuous movement [33]. Pink muscle, in pink color, is also called the intermediate muscle because of the intermediate region between red and white muscle [35,36]. Pink muscle fibers are recruited at an intermediate speed between white and red muscle fibers [34]. Although there is a functional separation between red and white muscle fibers as the slow and fast fibers, this separation of roles is not absolute.

### 3.2. Fish Muscle Alignment

Fish muscle is generally similar to terrestrial animal muscle in their composition and function [37]. The most distinguishable difference in striated muscle structure between fish and common livestock is their separation of different types of muscle fiber [31]. The edible part of the fish fillet could be further divided into myotomes (muscle fiber) arranged alternately with myocommata in a W-shaped pattern or connective tissue [35,38]. Another distinctive difference between fish muscles against terrestrial animals is the anatomical separation of fish muscles macroscopically into three groups: namely white muscle, red muscle, and intermediate pink muscle [35]. Overall, finfish have a lower collagen content than terrestrial animals [16]. The reason for the lower collagen content in finfish is that fish live in an environment that does not require as much support as terrestrial animals do [38]. Furthermore, the solubility of fish collagen is considerably higher than that of terrestrial mammals [39]. Lower collagen content in fish could potentially contribute to a softer fish fillet texture than a terrestrial animal fillet. Different from terrestrial animals, in which their muscle fibers are embedded in more complex hierarchical layers of connective tissue, fish lack the tendinous system that can connect muscle bundles to the skeleton [40,41]. Myocommata in fish can work similarly to the epimysia in terrestrial animals [35]. The absence of these collagenous fibers and tendons in fish makes them easier to be digested and absorbed [17]. Fibers in fish muscle have a diameter ranging from 10 to 100 μm and are shorter (the scale of millimeters) than those in terrestrial animals (usually a few centimeters) [35]. The muscle fibers are comprised of 1000 to 2000 myofibrils packed together into bundles, and each has a diameter of up to 5 μm. Myotomes in fish fillets consist of muscle fibers that run parallel to the longitudinal direction of the fish [16].

## 4. Processing Techniques to Produce Fish Analogue

Considering the complicated structures of fish protein, it is noteworthy that the preparation of fish fillet analogues requires more elaborate techniques to resemble the texture and appearance of fish meat when compared to the comminuted meat analogues production. Generally, two main strategies, namely the bottom-up strategy and top-down strategy, have been used to process plant-based meat analogues [42]. Bottom-up strategy assembles individual elements into a larger structure, even resembling the meaty hierarchical structure down to the nanoscale. However, the problems associated with this strategy include the cost of mass production and processing reliability [42]. On the other hand, a top-down approach imitates the structure of meat only on a much larger length scale, and could create anisotropic and fibrous products. It has a better scalability chance for economic application [42]. Nonetheless, the question associated with top-down strategy includes whether or not it could sufficiently mimic the actual meat products [42].

To date, very little successs has been had in producing fish analogues from plant-based proteins, but there are a handful of cases where novel processing techniques were developed to modulate plant proteins for other applications, such as nanofibers, edible films, meat analogues, and gels (Table 3). Those applications revealed the underlying mechanisms of plant protein behaviors under different processing conditions as well as their interactions with other compounds, which could be used to guide the process design of plant protein-based fish analogues. Each promising technique will then be extensively discussed in terms of its potential application for fish analogue production. Moreover, current successful commercial plant-based meat products focus on comminuted meat products such as sausage and burgers, where a list of functional ingredients was used. According to Table 4, the functionalities of these ingredients are binding agents and stabilizers, which aim to modulate the texture and mouthfeel of the plant-based products. More recently, DSM^®^ developed and commercialized a fish flavor [43], which enables manufacturers to formulate authentic, tasty, and enticing vegan seafood products.

### 4.1. Electrospinning

Electrospinning (Figure 3a) is a robust, cost-effective, and rapid technique for producing structured fibrous food with interconnected pores in sub-micron range [53]. Some unique features of electrospun nanofibers include high surface-to-volume ratio, tunable porosity, flexibility in conforming to different sizes and shapes, fiber morphology, and mechanical strength [53]. Recently, the application of electrospinning has been extended to processing plant-based proteins as meat analogues. Plant proteins, such as soy proteins and pea proteins, contain a mixture of various protein fractions with molecular weights ranging from 200 to 600 kDa [54]. In the past, collagen and gelatin have been widely studied for electrospinning due to their naturally fibrous structure [54]. Globular proteins, such as soy protein, have to go through an unfolding and denaturation process prior to electrospinning [54]. Recent work reported that a combined treatment of alkaline and thermal at a pH above 4.5 could help to unfold and solubilize soy proteins via exposing hydrophobic and sulfhydryl groups [54].

In fish analogue processing, it is critical to revitalizing the unique microstructure to ensure the proper texture. Therefore, precise alignment of fibrous proteins is necessary through electrospinning to achieve the highly ordered W-shape patterns to mimic the alignment of myotomes. An approach has been reported that can simultaneously combine electrospinning and electro-spraying to process two heterogeneous materials [57], demonstrating the potential of producing fish analogues through a precise bottom-up assembly of hierarchical microstructure. However, more research needs to be performed to improve the throughput and lower the production cost. In addition, most of the current solvents to unfold and solubilize plant proteins are toxic, such as hexafluoroisopropanol (HFIP) and trifluoroethanol (TFE) [54], more food-grade alternative solvents need to be explored.

### 4.2. Wet Spinning

Similar to electrospinning, wet spinning requires pumping polymer solution through a spinneret that is usually immersed in an acid bath, where the spun filaments can solidify, and then the solidified fibers can be collected and form a continuous tow and rope [58]. Wet spinning (Figure 3b) is considered a mild process because it operates at lower temperatures, which could be more feasible for processing food products.

Due to its capability of producing a variety of fiber cross-sectional structures, wet spinning also presents great potential in structuring plant proteins. A pioneering study used wet spinning to process soy protein to produce edible films, and the produced fibrous soy protein film exhibited strong tensile strength at 3.9 MPa and 55 MPa of Young’s modulus [45]. Although the number could vary significantly by fish species, the Young’s modulus of the fish fillet is much smaller than the edible film and meat. According to recent work, the magnitude of Young’s modulus in fish is around 10 KPa or 0.01 MPa [59], compared to 55 MPa in protein film and 0.1 MPa in beef [60]. Additionally, fish fillet has a 6 KPa to 10 KPa of loss modulus, indicating a mild weak network structure.

Boyer patented the wet spinning of proteins for the application of meat analogues in 1954 [42]. They extruded the solubilized soy protein through a spinneret and obtained stretched protein filaments with a thickness in the order of 20 μm [61]. The structure of a wet-spinned protein can be modulated by controlling the solidification process. Depending on the manipulation of the continuous phase and dispersed phase, capillary-filled gels and fiber-filled gels can be obtained [62]. The modulation of protein structure in wet-spinning is on a larger scale, where the secondary structure and tertiary structure did not shift significantly after processing [59]. The restricted capability of controlling the secondary and tertiary structure of proteins may limit the application of wet spinning in producing fish analogues with desirable textures.

### 4.3. 3D-Printing

Three-dimensional printing (3D printing) can be printed in micron size and has been gaining increasing attention in the food industry as a novel processing technique to produce novel food, such as recombined meat products, in the last few decades [63]. In addition to animal proteins, a wide range of plant-based materials have also been extensively studied for 3D printing, such as soy protein, pea protein, and wheat protein [52]. Many successes were achieved using soy proteins to produce a variety of meat analogues [64,65]. Although there has been a lot of work completed, challenges remain for 3D printing using plant-based proteins, especially regarding the structural reconstructive to obtain proper texture and to deliver the fine resolution of printed constructs with an appreciable mouthfeel. It was found that plant proteins without additives or pretreatments lack the capability needed to form a proper 3D structure [66].

A few strategies have been proposed to finely tune the 3D structure and viscoelastic properties of 3D-printed plant-based foods, which could be summarized in three categories: (1) process control, (2) co-printing with other biopolymers, and (3) use of enzymes. Process control is the most facile approach to modulate the structural and rheological properties of plant proteins during 3D printing. For example, a heating–cooling procedure has been found to improve the printing quality and accuracy of soy-based products [64]. In regard to co-printing, surface-active biopolymers (e.g., OSA-modified starch) have been found to significantly improve the shear-thinning behavior, and an increase in biosurfactants concentration led to an increase in viscosity recovery, yield stress, and elastic modulus [65]. It was also reported that monodispersed biosurfactants with a greater radius of gyration provide stronger pseudoplasticity [65]. Other than surface-active polymers, xanthan gum, gelatin, sodium alginate, and sodium chloride were found to strengthen the 3D printing structure for soy protein [48], yet more mechanistic insights are still needed. Enzymatic treatments could also be useful in modulating plant-based proteins during 3D printing. Transglutaminase (TGase), a very popular cold-set binder, could be used in restructured food and in plant-based products (e.g., pea, soy, wheat) [67], to help bind different meat analogues cohesively [68]. To date, very few approaches have explored the possibility of using 3D printing to produce fish analogues from plant-based proteins. It is expected that a combination of those above-mentioned strategies, including precise processing, the addition of synergistic biopolymers, and the application of enzymatic treatments, will enable fine control of the printed protein microstructure to achieve the unique mouthfeel of fish products. Examples of 3D-printed plant protein products can be found in Table 5.

### 4.4. Extrusion

High moisture extrusion (>40%) is frequently used for producing soy-based meat analogs [47], as it has many advantages, such as higher energy efficiency and less production expansion [72]. However, extrusion is an extremely complex processing technique that involves a variety of chemical and physical evolutions simultaneously [73]. Therefore, temperature control during the extrusion processing is the key to achieving desirable protein structure and texture. For example, in the production of soy protein-based meat analogs, the number of temperature zones can vary from four to nine in order to precisely control the plant protein structure [46,47,74]. The highest cooking temperature zone, which has the most profound impact on protein texturization, is usually set in the middle or rear-end of the barrel [75,76]. More importantly, recent work found that a high processing temperature could produce a V-shape structure, which attributes to the laminar flow in the die section [46] and can potentially mimic the mouthfeel of fish muscle. In addition to temperature control, protein and moisture content are also critical in determining the product properties. Using soy protein as an example, high moisture content (~77%) leads to a juicer and softer meat analogue, which is perceived as less fibrous but mushy mouthfeel in sensory evaluation [47]. On the other hand, higher protein content in plant materials allows the formation of more chemical bonds and contributes to the formation of a more chewy texture during extrusion [77]. Modulating texture profile by extrusion parameters to mimic fish fillet is yet to be explored, but current findings from meat analogue may suggest an extrusion process with high temperature, low protein concentration, and high moisture content could be preferable.

Other than process control, a list of functional ingredients could be incorporated to further improve the product quality, including texture profile, flavor profile, and color development [78]. Leghemoglobin has been successfully applied by Impossible Foods^TM^ to provide a meaty flavor and a red-to-brown color change upon cooking [52,78]. Since fish fillet has a distinctive color and flavor when compared to meat products, alternative additives will be needed to imitate fish products. It has been found that norbixin could potentially help to mimic the yellowish color in salmon to give a red-yellow color [79]. Compounds such as furans and thiophenes can provide a strong meaty aroma at low threshold values [80]. The addition of oil during the processing can contribute to juicier, tenderer meat analogs, and helps flavor release [81]. Common practice is to combine solid oil (e.g., coconut oil, palm oil) with liquid oil (e.g., sunflower oil, canola oil) to improve mouthfeel and juiciness during meat analogs production [50]. Examples of plant-based meat analogues from high moisture extrusion can be found in the Table 5.

## 5. Conclusions

The concept of developing fish and seafood analogues based on plant protein sources has been proposed recently [29]. However, there are still very limited successful applications of plant-based seafood analogues. This study has explored the nutritional properties of plant-based materials and has compared their nutritional properties, noticeably their protein quality with fish as it is nowadays being regarded as a novel source for plants compared to terrestrial meat. A combination of plant proteins from different sources makes it possible to achieve a very high PDCAAS comparable to fish protein. The difference in fish muscle and terrestrial animal muscle structure, both macroscopically and microscopically, has been emphasized. The main difference in fish muscles is their unique W-shaped muscle structure and the separation of myotome and myocommata. The fact that fish are not frequently consumed in the forms of value-added products such as ham and sausage renders a big challenge in producing fish analogues. However, various processing techniques have been proposed. Some techniques show potential in producing fish analogues that can emulate the unique W-shaped muscle structure, such as electrospinning and 3D printing.

Challenges and hurdles: Although there have been successful applications of plant-based beef burgers and sausage, most applications are mainly minced meat, lacking the characteristic appearance of whole-cut meat fillet. This could be due to the processing techniques for present plant-based meat analogues, that is, extrusion. This technology is applied generally to produce a product that only gives a consistent look. Despite the absence of the characteristic appearance of marine fish fillets, mimicking the texture of fish fillets is another factor that hinders the mass production of plant-based protein. Moreover, the connective tissue in cooked fish fillets could be destabilized, contributing to breaking the W-shaped structure upon mastication. Imitation of this behavior is yet another critical challenge. There have been some emerging processing technologies recently, such as wet spinning and electrospinning, which show the potential of producing fish analogue with their characteristic appearance and texture. However, these technologies are mainly used for scientific research, and the cost of these technologies is too high for mass production. Although plant-proteins are widely recognized as a sustainable alternative to animal proteins, their negative environmental impact during production also needs to be properly handled. For example, typical soybean has a protein content of 30 to 40% [8], and a high amount of waste stream could be produced during the soybean protein extraction procedure [82], yet more efficient soybean protein extraction methods are still underway.

Future perspectives: Traditional processing techniques to produce meat analogues limit the development of meat analogues with a more refined texture. In this light, more novel processing techniques that focus on producing analogues more microscopically have been developed. Comparing all the current processing techniques, extrusion is the predominately used technique in the current industry, while 3D-printing and electrospinning have shown the prominent potential of mimicking fish muscle structure as bottom-up approaches. However, their applicability for producing fish analogues needs further investigation. Furthermore, whether these techniques could work in combination also needs to be proved. Although it is possible to produce a fish analogue with the typical fish fillet appearance with these novel processing methods, their behavior upon cooking is another critical problem affecting their texture when consumed, and more studies need to be conducted on this subject. Finally, more studies are required to study the emulation of fish innate flavor.

## Figures and Tables

**Figure 1 foods-12-00614-f001:**
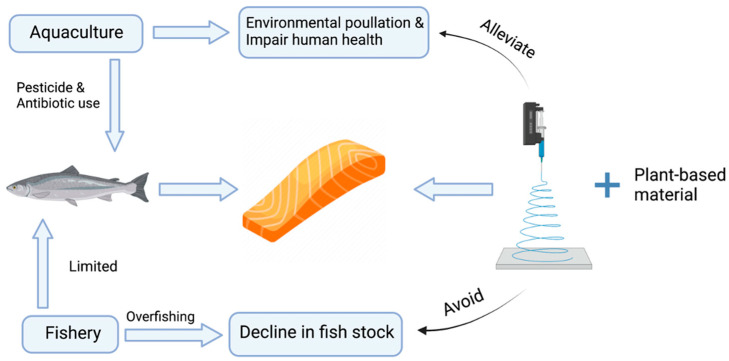
Rationale for developing plant-based fish analogues and the negative impact of fishery and aquaculture.

**Figure 2 foods-12-00614-f002:**
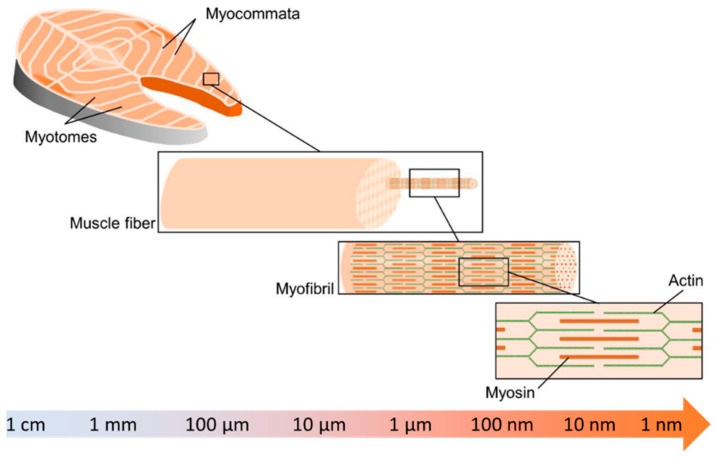
A schematic diagram indicating the hierarchical structures of fish muscle structure (Adapted from Kazir and others with permission) [29].

**Figure 3 foods-12-00614-f003:**
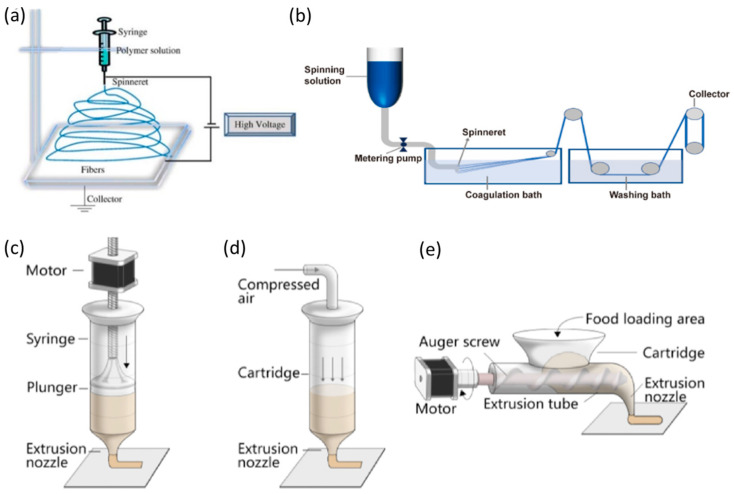
(**a**) Standard vertical setups of electrospinning process setup reprinted from [53] with permission. (**b**) Schematic description of a wet-spinning process for plant proteins, reprinted from [55]. (**c**) Syringe-based type extrusion, (**d**) air-pressure-driven extrusion, (**e**) screw-based type extrusion. (**c**–**e**) reprinted from Sun and others with permission [56].

**Table 1 foods-12-00614-t001:** Moisture, protein, and fat content of selected fish samples.

Fish	Moisture (%)	Protein (%)	Fat (%)	References
Cod, Atlantic (Gadus morhua)	80.8	18.2	0.11	[12]
Atlantic salmon (Salmo salar)	65.6	19.6	14.4	[13]
Tuna, yellowfin (Thunnus albacares)	70.3–72.7	21.8–25.1	3.1–5.7	[14]
Tuna, skipjack (Katsuwonus pelamis)	66.5–72.3	20.72–27.98	3.47–5.73	[14]

**Table 3 foods-12-00614-t003:** A summary of processing techniques to obtain different plant protein-based products.

Processing Techniques	Raw Material	Products	References
Electrospinning	Soy protein isolate and alginate	Nanofibers	[44]
Wet spinning	Soy protein isolate	Edible films	[45]
Extrusion	Hemp and soy protein	Meat analogue	[46]
Extrusion	Soy and algae	Meat analogue	[47]
3D printing	Soy protein isolate and xanthan gum	Gels	[48]
Extrusion	Pea protein and out fiber	Fibrous Meat analogue	[49]

**Table 4 foods-12-00614-t004:** Common ingredients used in plant-based products and functionalities.

Ingredients	Functionality	Example of Food	Reference
Polysaccharide (Carrageenan; Methylcellulose)	Texture enhancement; thickening agent	Meat analogues	[50]
Oleosomes	Emulsification/Stabilization	Meat analogues	[51]
Plant fat (Coconut oil; Canola oil; Sunflower oil)	Texture (juiciness) enhancement	Plant-based burger	[50]
Leghemoglobin	Meat flavor	Plant-based meat	[52]
Maxavor^®^ Fish YE	Fish flavor	Plant-based seafood	[43]

**Table 5 foods-12-00614-t005:** Plant-based meat analogues with different processing techniques.

Techniques	Plant Protein Sources	Processing Parameters	Images	Ref.
High moisture extrusion	Soy protein isolate	Extrusion temperature = 95 °C, Screw speed = 600 rpm, Moisture content = 57%	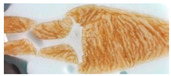	[46]
3D printing	Pea protein isolate	Nozzle diameter = 1.54 mm, Extrusion speed = 15 mm/s	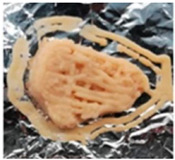	[69]
High moisture extrusion	Soy protein isolate	Extrusion temperature = 124 °C, Screw speed = 250 rpm, Water content = 50%	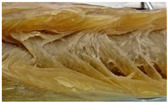	[70]
3D printing	Textured soybean protein	Nozzle diameter = 0.8 mm, Extrusion speed = 20 mm/s	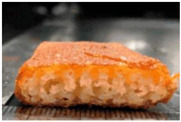	[71]

## Data Availability

The data are available from the corresponding author.

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
