# Peer review of "Production of Fish Analogues from Plant Proteins: Potential Strategies, Challenges, and Outlook"

_foods, 2023, doi:10.3390/foods12030614_

Round 1

Reviewer 1 Report

The manuscript foods-2170791 describe technologies for production of plant-based fish analogs. The review  has a first part describing the fish proteins and a second part related to the technologies for preparing fish protein analogs.

I do not consider the review particularly interesting and no new information are reported, in addition the introduction part and the rest of the manuscript are not well correlated to eachother.

Some comments:

1) in the introduction should be specified the different fishing strategies

2) Line 62-63 references should be added

3) the aminoacidic composition of the protein is not considered at all in the review, which represents a key point in the development of analogues

4) nutritional values of the analogue should be compared with the fish protein

5) it has to be considered that growing and harvesting terrestrail plants have also an economical and environmental impact

Reviewer 2 Report

The review manuscript by Zhong et al critically overviews the various prospects of fish analogue. The manuscript summarize the currently available literatures on the feasibility and production strategy, and put forwards some interesting recommendations. The hypothesis is properly stated. The manuscript is written in easy-to-understand and read language.

However, to further improve it, I have the following suggestions as-

                 i.          L13: is overfishing the only reason behind the depletion of fish resources; I think pollution an other reasons are also contributing it.

               ii.          L19-21; please include some salient recommendations or conclusion

             iii.          L107: space between species were

             iv.          L117: Fish structure or I think Fish muscle structure more appropriate

               v.          L207: plz check the reference style

             vi.          Please the adding section on the firms who are in the field and developed the product. It will improve the readership and quality of the manuscript, if possible with their common name.

            vii.          A short section of various ingredients used will improve the quality.

          viii.          Plz may add one new section on Future prospects or add Future prospects and conclusion

Thanks for giving me the opportunity to read your work. 

Reviewer 3 Report

The manuscript is interesting, but I have some remarks mentioned below

- Please rewrite and organize the abstract according to the following context:

A short introduction, hypothesis (aim) of the study, methods, the most important quantitative results, a general conclusion, and future prospective

 - The authors should include some information about applied methods at the end of the introduction. The information should contain data sources, search terms and search strategies, selection criteria, the number of studies screened and the number of studies included, etc. Please provide this info to the readers.

- Please provide some pictures of both the processing techniques mentioned and plant protein-based products produced from them.

 - Conclusions section, please highlight the future standpoint well.

- Some references should be updated.

-  Manuscript has grammatical errors, please check.

Round 2

Reviewer 1 Report

the authors have improved the manuscript